# Length of Hospital Stay, Hospitalization Costs, and Their Drivers in Adults with Diabetes in the Romanian Public Hospital System

**DOI:** 10.3390/ijerph191610035

**Published:** 2022-08-14

**Authors:** Cornelia Bala, Adriana Rusu, Dana Ciobanu, Gabriela Roman

**Affiliations:** 1Department of Diabetes and Nutrition Diseases, Iuliu Hatieganu University of Medicine and Pharmacy, 400012 Cluj-Napoca, Romania; 2Diabetes Centre, Emergency Clinical County Hospital, 400006 Cluj-Napoca, Romania

**Keywords:** direct costs, hospitalization, length of hospital stay, diabetes mellitus

## Abstract

The aim of this analysis was to assess the costs associated with the hospitalizations of persons with diabetes in a Romanian public hospital. We performed a retrospective “top-down” cost analysis of all adult patients discharged from a tertiary care hospital with an ICD-10 primary or secondary code of diabetes mellitus (type 1, type 2, or specific forms) between 1 January 2015 and 31 December 2018. All costs were adjusted with the annual inflation rates and converted to EUR. We included 16,868 patients with diabetes and 28,055 episodes of hospitalization. The total adjusted hospitalization cost in the analyzed period was EUR 26,418,126.8 and the adjusted median cost/episode of hospitalization was EUR 596.5. The mean length of a hospital stay/episode was 7.3 days. In the multivariate regression analysis, higher adjusted average costs/episodes of hospitalization and longer lengths of hospital stays were associated with increasing age, the presence of cardiovascular diseases, chronic kidney disease, and foot ulcerations. Moreover, a significant association between the average cost/episode of hospitalization and the length of hospital stay was observed (β = 0.704, *p* < 0.001). This study shows the burden on Romanian public hospitals of inpatient diabetes care and the main drivers of the costs.

## 1. Introduction

Diabetes is a serious chronic condition with major medical, social, and economic impacts, estimated to have affected 537 million adults aged 20 to 79 worldwide in 2021 and projected to increase to 783 million by 2045 [1]. The worldwide prevalence of diabetes has reached 10.5%, while the prevalence of diabetes in Romania was estimated at 8.4% [2]. An epidemiological study reported an even higher prevalence of diabetes (12.4%) in the Romanian adult population, positioning Romania among the European countries with the highest diabetes prevalence [3].

Diabetes is associated with decreased quality of life, reduced life expectancy, and increased mortality related to both acute and chronic complications [1]. Beyond the negative impact of diabetes mentioned above, we must also consider the economic burden of this disease on the healthcare systems. Worldwide, the economic impact of diabetes has been on the rise in the past two decades [1]. The International Diabetes Federation (IDF) Atlas underlined the significant economic impacts of diabetes on health budgets around the globe starting with its first edition in 2000 [3]. Using recent epidemiological data and a more sophisticated economic analysis, the latest edition of IDF Diabetes Atlas reported the estimates of diabetes-related health expenditures at national, regional, and global levels, which have followed an ascending trend, growing from USD 232 billion in 2007 to an estimated USD 966 billion worldwide in 2021 [1]. It is expected that the economic impact of diabetes will increase by 66.4% in 2030, reaching USD 1.03 trillion, assuming that diabetes prevalence and mean costs per person will remain constant [1].

The European region has spent a lower percentage of health costs for diabetes care (8.6%) compared to South and Central America, the Middle East, and North Africa, where 18.4% and 16.6% were spent, respectively [1]. On the other hand, out of the top ten countries with the highest health expenditures on diabetes per person, nine countries are from Europe [1]. The highest diabetes-related health expenditures were estimated for the United States of America (US) at USD 379.5 billion, while the highest yearly expenditure per person was estimated for Switzerland at USD 112,828 followed by the US at USD 11,779 [1]. Furthermore, it was estimated that the costs have increased by 26% in the past years due to higher direct and indirect costs per person, and the increasing prevalence of diabetes [4]. Variations in costs associated with the care of persons with diabetes have been correlated with age and type of said illness. Larger health spending was observed with older ages mainly due to an increase in diabetes complications [1]. The overall cost/year of hospitalizations for persons with diabetes was GBP 3.0 billion higher compared to people without it, with type 1 diabetes having three times higher costs as compared to type 2 diabetes [5].

The mean diabetes-related expenditure per person with diabetes in Romania was estimated at USD 930.2 per year, as reported in the 10th edition of the IDF Atlas [1]. An analysis of the outpatient diabetes costs in Romania from 2000 to 2017 reported that diabetes accounted for 21.3% of the total funds allocated for national healthcare programs, with an average growth rate of 25.4% during this period [6]. When assessing the hospitalization costs of treating complications of diabetes in Romanian patients, only one recent study is available, which reported that the cost was 40% higher in patients presenting with lower limb ulcerations and amputations compared to those not presenting with this complication [7]. Even at additional costs, optimal treatment of people with diabetes is the main solution for preventing acute and chronic complications, which would result in an even higher financial burden.

Although the economic burden of diabetes has been reported in several countries, the costs of in-hospital medical care for persons with diabetes remain unknown/not publicly available for Romania. Since the use of pharmacoeconomic evidence is increasing in Romania [8], an analysis of actual costs of the in-hospital care of patients with diabetes is expected to provide support for policymakers in financial decision-making and better resource allocation.

In this context, we hypothesized that in Romania, as in other countries, the costs of hospitalization of patients with diabetes are high and hospital care of these patients accounts for a high proportion of the hospital expenses. In the research presented here, we aimed to calculate the costs associated with the hospitalization of adults with diabetes and to identify the main drivers of these costs.

## 2. Materials and Methods

### 2.1. Data Source, Study Sample, and Data Analyzed

In this retrospective database analysis, we used data obtained from the electronic system of the Emergency Clinical County Hospital Cluj-Napoca, Romania, a tertiary care hospital covering the population of the Transylvania region. This system contains information on dates of admission and discharge, demographics, laboratory investigations, and therapy administered during hospitalization, admission, and discharge diagnoses coded according to the International Classification of Disease Tenth Revision (ICD-10). As we did not have access to the database of secondary care hospitals in the region, we chose to analyze the data from the Emergency Clinical County Hospital Cluj-Napoca where adult patients from the Transylvania region are referred for health issues that cannot be addressed in the secondary care hospitals (extremely poor glycemic control, starting insulin pump therapy in type 1 diabetes, severe neuropathy or peripheral arterial disease). Moreover, the Emergency Clinical Hospital Cluj works as a secondary care hospital for adult patients from Cluj-Napoca and the surrounding areas. Thus, analyzing the data from this facility would provide a good overview of diabetes-associated costs in both secondary and tertiary care hospitals.

We included data of adult patients (≥18 years of age) with hospitalization between 1 January 2015 and 31 December 2018 and with a primary or secondary diagnosis at the discharge of type 1, type 2, or specific forms of diabetes (ICD-10 codes E10, E11, E13, E14, O24). The full list of ICD codes used is provided in the Appendix A. For all patients included, we analyzed data on age, sex, number of episodes of hospitalization, length of hospitalization for each episode, total costs/episode of hospitalization, and diabetes-specific acute and chronic complications.

### 2.2. Ethical Considerations

The study protocol was approved by the Ethics Committee of the Emergency Clinical County Hospital Cluj-Napoca, Romania. Due to the retrospective database analysis, informed consent was waived. The data were exported by the IT service provider of the institution and analyzed without any personal identifiers. Each participant received a code that allowed identification in the database.

### 2.3. Hospitalization Costs

Patient level data per episode of hospitalization for direct inpatient care costs were extracted from the electronic system of the Emergency Clinical County Hospital Cluj-Napoca. An episode of hospitalization was defined as the time between the patient’s admission to the hospital until discharge. Readmission was treated as a new episode of hospitalization.

Data used for cost/episode of hospitalization were the ones available at discharge as listed in the hospital electronic system irrespective of the reimbursement status. The costing methodology used was a “top-down” one; hospitals in Romania do not record unit costs per service at the patient level; the average cost per service resulting from total costs divided by the number of services performed was used [9,10,11]. Patient level data are available only for drugs used and length of stay [11].

The direct inpatient care costs of persons with diabetes used in this analysis consisted of a sum of:Cost of beds, which was calculated as the number of days of hospitalization multiplied by the cost/day of hospitalization. The costs per day of hospitalization are in a fixed amount;Cost of food, which was calculated as the number of days of hospitalization multiplied by the daily food allowance (fixed amount);Cost of drugs used during the hospitalization;Cost of medical supplies and medical devices;Cost of services: laboratory and imagistic investigations (X-ray, ultrasonography, CT scans, MRI scans), surgical procedures.

All cost data were recorded in the electronic system of the Emergency Clinical Hospital Cluj in the Romanian Leu. As previously described [6], to account for the inflation during the analyzed period, costs were adjusted with the annual inflation rates calculated by the Romanian National Statistics Institute using the Consumer Price Index (CPI, INS CPI-Annual Data Series available online: http://www.insse.ro/cms/ro/content/ipc–serie-de-date-anuala; accessed on 21 June 2020) and converted to EUR, taking into account the average exchange rate of the Romanian National Bank, for each year of the period 2015–2018 (National Bank of Romania—Exchange Rates: Monthly, Quarterly and Annual Averages (available online: https://www.bnr.ro/Exchange-Rates--3727.aspx, accessed on 21 June 2020). The base year chosen was 2018.

### 2.4. Romanian Healthcare System Context

A national healthcare plan for diabetes covered by the Social Health Insurance system is in place in Romania. It covers all direct costs of diabetes care for both insured and uninsured patients with no registered income. Moreover, based on the diagnosis of diabetes, patients with no registered income are entitled to health insurance that covers free of charge all direct costs associated with their medical care in the public healthcare system for any disease. Hypoglycemic drugs are usually prescribed in outpatient settings by physicians specializing in diabetes, nutrition, and metabolic diseases. The frequency of specialist appointments varies according to the glycemic control and the type of hypoglycemic medication prescribed (every 3 months for those treated with insulin and newer drugs and every 6 months for those treated with metformin and sulfonylureas). For acute and chronic complications or poor glycemic control that cannot be addressed in the outpatient setting, persons with diabetes are referred to secondary and tertiary care hospitals.

Costs and reimbursement rates of public hospitals in Romania are based on the Australian Diagnosis Related Groups (DRG) system with a mix of prospective and retrospective reimbursement systems, which was previously described [12]. Thus, for the public healthcare system, the Ministry of Health and the National Health Insurance House establish fixed costs/day of hospitalization for bed usage, which varies according to the type of hospital and hospital department based on the severity of cases in the previous year. Moreover, the daily food allowance is a fixed amount. In addition, the Ministry of Health regulates the costs of reimbursed drugs and of drugs purchased by the hospitals by direct negotiation with marketing authorization holders and by imposing a claw-back tax with the aim to control drug expenditures [13].

### 2.5. Statistical Analysis

No sample size calculation was performed, and all episodes coded as hospitalizations within the target period were included in the analysis.

The number of episodes of hospitalization was reported overall and per participant. Total CPI-adjusted costs for hospitalizations and CPI-adjusted average cost per episode of hospitalization were reported for the whole sample and stratified per year, age group, sex, type of diabetes, and type of diabetes acute and chronic complication. The yearly CPI-adjusted costs for hospitalization of participants with diabetes were expressed as a percentage of yearly current expenses of the hospital and the yearly total number of episodes of hospitalization for participants with diabetes as a percentage of the total episodes of hospitalization in the hospital in the respective year. The figures for the entire hospital are publicly available online (https://scjucluj.ro/pdf/intpublic/00.%20Analiza_Spitalul%20Clinic%20Judetean%20de%20urgenta%20Cluj_10_12_2020.pdf, accessed on 1 November 2021).

Statistical analysis was performed using SPSS version 26 (SPSS, Inc., Chicago, IL, USA). Data are presented as mean (95% confidence interval [CI]) and number (percentage). The comparison between study groups was performed using the t-test, independent sample Kruskal–Wallis, ANOVA, and chi-square tests. Multivariate linear regression with age, sex, diabetes type, and acute and chronic complications as predictors was used to investigate the main predictors of median CPI-adjusted costs per episode of hospitalization. A *p*-value < 0.05 was considered statistically significant.

## 3. Results

### 3.1. Study Sample

This analysis included 16,868 persons with diabetes for whom were recorded 28,055 episodes of hospitalization between 1 January 2015 and 31 December 2018. The number of episodes of hospitalization according to age, sex, type of diabetes, and year are presented in Table 1. Most hospitalization costs were in patients with type 2 diabetes and middle-aged adults and the elderly. A total of 43.0% of the episodes were in patients with CVD, while 2.6% of the hospitalization episodes were for acute diabetes complications.

### 3.2. Length of Hospital Stay

Overall, the mean length of hospital stay was 7.3 days and was significantly longer in patients with type 2 diabetes and specific forms of diabetes as compared to type 1 diabetes (*p* < 0.001). It also increased with age (from 5.5 days in those 18 to 40 years of age, 6.7 days in those 40 to 65 years of age, and 8.0 in those ≥65 years of age, *p* < 0.001) and was higher in women than in men (7.5 days vs. 7.2 days, *p* = 0.002). With regards to diabetes chronic complications, the length of hospitalization was longer in those with diabetic neuropathy, chronic kidney disease, foot ulcerations, and CVD (*p* < 0.05 for all). Patients with acute diabetes complications had an average duration of hospitalization significantly lower as compared to those without acute complications (*p* < 0.001).

The length of hospital stay increased from 2015 reaching a maximum in 2016 and decreased thereafter with a minimum duration in 2018 (*p* = 0.002). A similar evolution was seen in the length of hospital stay in patients with type 2 diabetes, women, middle-aged adults, and patients with diabetic retinopathy (*p* < 0.05 for all). No change over time was observed in type 1 diabetes, specific forms of diabetes, men, young adults, elderly, patients with diabetic neuropathy, chronic kidney disease, foot ulcerations, and CVD (Table 2).

Predictors of a longer length of hospital stay were increasing age, a diagnosis of chronic kidney disease, foot ulcerations, and CVD. Diabetic retinopathy and acute diabetes complications were predictors of a shorter hospital stay. Sex, diabetes type, and diabetic neuropathy were not associated with the length of hospital stay (Table 4).

### 3.3. Cost of Hospitalization during the Analyzed Period

The total CPI-adjusted cost of hospitalizations of persons with diabetes in the analyzed period was EUR 26,418,126.8. The total costs per year had an increasing trend in the period 2015–2017 in parallel with an increasing number of episodes of hospitalizations. In 2018, the total costs decreased, while the highest number of episodes of hospitalization in the studied period was registered (7408 episodes) (Figure 1). The yearly total current expenses of the hospital for persons with and without diabetes were EUR 52,697,709.9 for 58,986 episodes of hospitalizations in 2015, EUR 62,191,672.49 for 57,951 episodes of hospitalizations in 2016, EUR 65,814,810.7 for 56,998 episodes of hospitalizations in 2017, and EUR 78,951,326.96 for 58,215 episodes of hospitalizations in 2018. Reported to these, the expenses for the hospitalizations of persons with diabetes represented 12.3% in 2015, 10.5% in 2016, 10.6% in 2017, and 8.1% in 2018. At the same time, the episodes of hospitalization for persons with diabetes represented 11.6% in 2015, 11.7% in 2016, 12.3% in 2017, and 12.7% in 2018.

The CPI-adjusted median cost per episode of hospitalization was EUR 596.5. It registered a maximum of EUR 621.9 in 2016 and decreased steadily thereafter to EUR 577.4 in 2018. Similar trends were observed for the adjusted costs per episode of hospitalization for type 2 diabetes and specific forms of diabetes, all age groups, chronic microvascular complications, and hospitalizations for cardiovascular diseases. No statistically significant difference was observed between studied years in participants with foot ulcerations and hospitalizations for acute diabetes complications (Figure 1, Table 3).

Overall, lower costs per episode of hospitalization were observed for episodes of acute diabetes complications, diabetic retinopathy, diabetic neuropathy (*p* < 0.001 for all), and higher costs for CVD (*p* < 0.001), foot ulcerations (*p* < 0.001), and chronic kidney disease (*p* = 0.010). At all-time points, the CPI-adjusted median costs per episode of hospitalization were significantly higher in those with type 2 diabetes and specific forms of diabetes as compared to type 1 diabetes, in those over 65 years of age as compared to younger ones, and those with CVD, foot ulcerations, and chronic kidney disease as compared to those without (*p* < 0.001 for all). By sex, the adjusted median costs per episode of hospitalization were significantly higher in women than in men only in 2016 and 2017 (*p* < 0.001). Lower costs were observed for episodes of hospitalization for acute diabetes complications than for hospitalization episodes for other ICD codes for the years 2015, 2016, and 2018 (Table 3).

### 3.4. Drivers of Hospitalization Costs

As most of the participants had multiple diabetes chronic complications, we assessed the predictors for CPI-adjusted costs per episode of hospitalization. Increasing age, male sex, CVD, chronic kidney disease, and foot ulcerations were associated with higher adjusted costs per episode of hospitalization. Acute diabetes complications, diabetic retinopathy, and diabetic neuropathy were predictors of lower costs of hospitalization. Diabetes type did not contribute to the multiple regression model (Table 4).

As the length of hospital stay varied significantly with all variables included in the multiple regression analysis, we performed a separate univariate regression with the length of hospital stay as the predictor and cost per episode of hospitalization as the dependent variable. The results showed a significant association between these parameters (standardized β coefficient = 0.704, *p* < 0.001).

## 4. Discussion

By analyzing data from a tertiary care hospital, we showed—for the first time for Romania—the costs associated with the public hospital care of persons with diabetes and their main drivers. Considering the projected increase in diabetes prevalence and the burden that diabetes already represents on the Romanian healthcare system, this study allows the identification of factors that may be targeted to prevent hospitalization and, thus, reduce the costs of care for these persons. The total costs for the analyzed period were substantial, exceeding EUR 26 million and the median cost of an episode of hospitalization in a person with diabetes was EUR 596.5, regardless of age, complications, and diabetes type.

High costs of diabetes care have been reported worldwide. The American Diabetes Association reported a cost of USD 237 billion for diabetes care in the US in 2017 (including USD 237 billion in direct medical costs) [4], increasing from USD 116 billion in 2007 [14] and USD 176 billion in 2012 [15]. According to the IDF, in the European region, the direct costs of diabetes accounted for 8.6%% of the annual total health expenditures [1]. Our results complement the recently published data on the evolution of costs of diabetes management in outpatient settings in Romania, which showed an increasing trend between 2000 and 2017 [6]. This increase could not be explained by the increase in the patient number alone. In the studied period, the costs increased by 14 times while the number of persons with diabetes by 2.5 times [6]. In Romania, the costs of care for persons with diabetes in outpatient settings are covered by a National Health Program and by funds allocated by the Romanian National Health Insurance House. During the analyzed period, the main drivers of the increasing costs in outpatient care were the introduction of innovative drugs, the reimbursement of self-monitoring of blood glucose in insulin-treated persons, HbA1c, and insulin pumps in those with type 1 diabetes [6].

As compared to other countries, the average costs per episode of hospitalization were significantly lower in our country. In a systematic review that included 92 studies, the direct annual costs per patient ranged between EUR 1279 for Italy and EUR 6505 for France in established markets and from EUR 396 in Brazil to EUR 2906 in China for emerging markets. Hospitalization accounted for 42% of these costs in both emerging and established markets [16]. Our results may be explained by the lower inpatient tariff charges, lower costs of investigations during the hospital stay, lower wages of medical staff, and drug costs, which are regulated by the Ministry of Health.

In the research reported here, we observed that the total annual costs had an increasing trend between 2015 and 2017 in parallel with the increasing number of hospitalizations during this time and slightly decreased in 2018. This decrease occurred despite the increasing number of episodes of hospitalization and reflects the lower average cost per episode of hospitalization in this latter year mostly due to the lower duration of hospitalization during this time and a significantly higher percentage of acute complications (associated with a lower cost per episode of hospitalization in our sample). The length of hospital stay is one of the major drivers of the hospitalization costs in patients with diabetes [17]. In a recent study examining the costs of the hospitalizations of persons with diabetes in an Irish public hospital, Friel et al. [17] reported an incremental effect of the length of hospitalization on the costs; each additional day of hospital stay increased the costs by 3 percentage points for type 1 diabetes and by 2 percentage points for the costs for type 2 diabetes. In our analysis, we also observed a significant association between the length of hospital stay and costs per episode of hospitalization and a similar evolution over time for the length of stay and hospitalization costs overall and specific for persons with type 2 diabetes.

Diabetes chronic complications are preventable and diabetes management strategies aiming to modify the risk for these complications are interventions that may help reduce healthcare costs. Our results showed that in addition to the length of stay and age, diabetes chronic complications (CVD, foot ulcerations, chronic kidney disease) were key drivers of costs. Previous studies have also reported diabetes-related chronic complications and length of stay as the main drivers of inpatient medical care [17,18,19]. In a recent study examining the hospitalization costs of persons with diabetes in an Irish public hospital, age and diabetes chronic complications were important drivers of hospitalization costs [17]. In an analysis of hospital-based care that included 392,200 persons with type 2 diabetes in Sweden, Andersson et al. [19] also found that diabetes micro- and macrovascular complications were the main cost drivers. Similar results were also reported in Bulgaria where 54% and 41% of the in-hospital care were attributable to diabetes chronic macro- and microvascular complications in type 2 diabetes and type 1 diabetes, respectively [20].

Previous research performed elsewhere reported high costs for care of acute diabetes complications [21,22] and higher costs per episode of hospitalization for acute diabetes complications as compared to those without [23]. In an analysis performed by Javor et al. [23], it was shown that diabetes ketoacidosis triples the costs of hospital care for patients with type 1 diabetes in the US (USD 13,096 per hospitalization episode for diabetes ketoacidosis versus USD 4907 per hospitalization episode without ketoacidosis). In our study, we found lower costs and we can explain these costs by the lower lengths of stay of these patients as compared to patients without acute diabetes complications probably due to the lower frequency of associated chronic complications, including CVD, and associated comorbidities. As opposed to Javor et al. [23], we included in our analysis patients with type 1, type 2, and specific forms of diabetes. This may have influenced the costs due to a higher burden of CVD, chronic diabetes complications, and other comorbidities in those with type 2 diabetes who were not admitted for acute diabetes complications. It has been shown that the length of hospital stay and the cost of in-hospital care for type 2 diabetes is significantly higher compared to that of type 1 diabetes [17,24].

Although we attempted to provide a detailed analysis of inpatient medical costs, our analysis has limitations. First, only the costs of persons discharged with a diagnosis of diabetes were available for this analysis. Thus, a comparison with inpatient costs of care of persons without diabetes was not possible. Another limitation is that we did not have the costs split per medication and investigation. No data on diabetes history, glycemic control, and other sociodemographic variables of participants were included in the database provided. For example, previous studies showed that hospitalization costs were influenced by smoking habits, with 24 to 35% higher costs for current and former smokers as compared to never smokers [25]. The availability of the information on these risk factors may have provided deeper insight into the figures found. The single-center design is another limitation of the study. Although our hospital functions as both a secondary and tertiary care hospital, the costs depicted could overestimate the costs associated with diabetes care in other secondary care hospitals.

## 5. Conclusions

This study shows the burden on Romanian public hospitals of inpatient diabetes care and the main drivers of the costs and length of stay. Total hospitalization costs of persons with diabetes in the analyzed period were substantial and the main cost drivers were the length of stay and diabetes chronic complications. The information provided here will be useful for policymakers to support informed decisions on diabetes prevention and outpatient management strategies. Such strategies include multifactorial therapeutic interventions, patient therapeutic education, and antihyperglycemic medication with demonstrated cardio–renal–metabolic benefits, which have been shown to prevent diabetes chronic complications and related hospital admissions.

## Figures and Tables

**Figure 1 ijerph-19-10035-f001:**
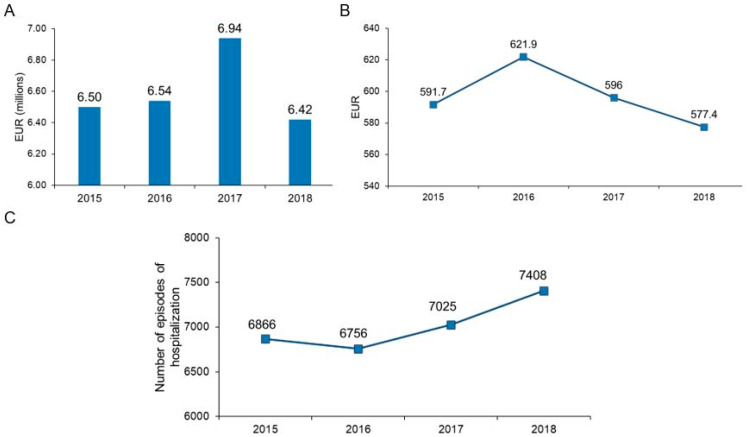
Evolution of total costs panel (**A**), median cost/episode of hospitalization panel (**B**), and the number of episodes of hospitalization in the studied period panel (**C**).

**Table 1 ijerph-19-10035-t001:** Description of the study sample (episodes of hospitalization).

	All Sample N = 28,055	2015 N = 6899	2016 N = 6756	2017 N = 7025	2018 N = 7408	*p*-Value for Trend
Type of diabetes, n (%)						
Type 1	1457 (6.2%)	386 (5.6%)	361 (5.3%)	343 (4.9%)	367 (5.0%)	0.164
Type 2	26,079 (93.0%)	6316 (92.0%)	6283 (93.0%)	6572 (93.6%)	6908 (93.3%)	0.002
Specific forms	519 (1.8%)	164 (2.4%)	112 (1.7%)	110 (1.6%)	133 (1.8%)	0.001
Age, years *	64.3 (64.2; 64.4)	63.9 (63.7; 64.2)	64.2 (63.9; 64.5)	64.3 (64.1; 64.6)	64.7 (64.4; 65.0)	0.001
Age groups, n (%)						<0.001
18–40 years	958 (3.4%)	226 (3.3%)	237 (3.5%)	250 (3.6%)	245 (3.3%)	
40–65 years	12,613 (45.0%)	3266 (47.6%)	3110 (46.0%)	3089 (44.0%)	3148 (42.5%)	
≥65 years	14,484 (51.6%)	3374 (49.1%)	3409 (50.5%)	3686 (52.5%)	4015 (54.2%)	
Women, n (%)	14,712 (52.4%)	3648 (53.1%)	3561 (52.7%)	3660 (52.1%)	3843 (51.9%)	0.427
Cases with diabetic retinopathy, n (%)	4171 (14.9%)	1093 (15.9%)	1026 (15.2%)	1029 (14.6%)	1023 (13.8%)	0.004
Cases with diabetic neuropathy, n (%)	5810 (20.7%)	1397 (21.8%)	1472 (21.8%)	1420 (20.2%)	1421 (19.2%)	<0.001
Cases with chronic kidney disease, n (%)	5074 (18.1%)	1218 (17.7%)	1231 (18.2%)	1256 (17.9%)	1369 (18.5%)	0.655
Cases with CVD, n (%)	12,074 (43.0%)	2910 (42.4%)	2806 (41.5%)	2991 (42.6%)	3367 (45.5%)	<0.001
Cases with foot ulcerations, n (%)	1102 (3.9%)	282 (4.1%)	275 (4.1%)	281 (4.0%)	264 (3.6%)	0.302
Cases with acute diabetes complications, n (%)	740 (2.6%)	140 (2.0%)	194 (2.9%)	172 (2.4%)	234 (3.2%)	<0.001

N/n (%), number (percentage) of episodes of hospitalizations; CVD, cardiovascular diseases. * Presented as mean (95% Confidence Interval).

**Table 2 ijerph-19-10035-t002:** Assessment of average length of stay * per episode of hospitalization.

	All Sample N = 28,055	2015 N = 6899	2016 N = 6756	2017 N = 7025	2018 N = 7408	*p*-Value for Trend
Overall	7.3 (7.2; 7.4)	7.4 (7.2; 7.6)	7.5 (7.4; 7.7)	7.3 (7.2; 7.5)	7.1 (6.9; 7.2)	0.002
Type of diabetes						
Type 1	5.9 (5.6; 6.1)	5.7 (5.2; 6.2)	6.1 (5.6; 6.6)	6.2 (5.6; 6.9)	5.4 (4.8; 6.0)	0.196
Type 2	7.4 (7.3; 7.5)	7.4 (7.3; 7.6)	7.6 (7.4; 7.8)	7.4 (7.2; 7.6)	7.2 (7.0; 7.3)	0.005
Specific forms	8.3 (6.7; 9.9)	9.9 (5.1; 14.8)	8.0 (6.7; 9.3)	8.2 (6.9; 9.6)	6.6 (5.8; 7.3)	0.485
*p*-value	<0.001	<0.001	<0.001	0.020	<0.001	
By sex						
Women	7.5 (7.4; 7.6)	7.4 (7.2; 7.6)	7.8 (7.5; 8.0)	7.5 (7.3; 7.8)	7.2 (7.0; 7.4)	0.003
Men	7.2 (7.0; 7.3)	7.4 (7.0; 7.7)	7.3 (7.1; 7.5)	7.1 (6.8; 7.4)	6.9 (6.7; 7.2)	0.129
*p*-value	0.001	0.925	0.005	0.035	0.125	
By age groups						
18–40 years	5.5 (5.2; 5.8)	5.0 (4.4; 5.7)	5.8 (5.2; 6.5)	5.9 (5.2; 6.6)	5.1 (4.6; 5.6)	0.108
40–65 years	6.7 (6.6; 6.9)	6.8 (6.6; 7.0)	7.0 (6.8; 7.3)	6.7 (6.5; 7.0)	6.4 (6.2; 6.7)	0.005
≥65 years	8.0 (7.8; 8.1)	8.1 (7.8; 8.5)	8.1 (7.9; 8.4)	8.0 (7.7; 8.3)	7.7 (7.5; 7.9)	0.061
*p*-value	<0.001	<0.001	<0.001	<0.001	<0.001	
Diabetic retinopathy						
Yes	5.1 (5.0; 5.3)	5.4 (5.1; 5.6)	5.3 (5.1; 5.6)	5.2 (4.9; 5.5)	4.7 (4.4; 5.0)	0.004
No	7.7 (7.6; 7.8)	7.8 (7.5; 8.0)	7.9 (7.8; 8.1)	7.7 (7.5; 7.9)	7.4 (7.3; 7.6)	0.006
*p*-value	<0.001	<0.001	<0.001	<0.001	<0.001	
Diabetic neuropathy						
Yes	7.4 (7.3; 7.5)	7.3 (6.7; 7.8)	7.0 (6.8; 7.3)	7.1 (6.8; 7.5)	6.9 (6.6; 7.2)	0.627
No	7.1 (6.9; 7.3)	7.4 (7.2; 7.6)	7.7 (7.5; 7.9)	7.4 (7.2; 7.6)	7.1 (6.9; 7.3)	0.001
*p*-value	0.006	0.531	0.001	0.304	0.330	
Chronic kidney disease						
Yes	8.1 (7.9; 8.3)	7.8 (7.5; 8.2)	8.2 (7.8; 8.7)	8.4 (8.0; 9.8)	8.0 (7.6; 8.3)	0.241
No	7.2 (7.1; 7.3)	7.3 (7.1; 7.5)	7.4 (7.2; 7.6)	7.1 (6.9; 7.3)	6.9 (6.7; 7.0)	0.001
*p*-value	<0.001	0.034	<0.001	<0.001	<0.001	
Foot ulcerations						
Yes	10.7 (10.2; 11.2)	11.6 (10.5; 12.6)	10.3 (9.3; 11.2)	10.1 (9.2; 11.0)	10.7 (9.7; 11.7)	0.164
No	7.2 (7.1; 7.3)	7.2 (7.0; 7.4)	7.4 (7.3; 7.6)	7.2 (7.0; 7.4)	6.9 (6.8; 7.1)	0.002
*p*-value	<0.001	<0.001	<0.001	<0.001	<0.001	
CVD						
Yes	8.0 (7.8; 8.1)	8.0 (7.6; 8.4)	8.1 (7.9; 8.4)	8.0 (7.7; 8.3)	7.8 (7.6; 8.1)	0.492
No	6.8 (6.7; 6.9)	6.9 (6.7; 7.2)	7.1 (6.9; 7.3)	6.9 (6.6; 7.1)	6.4 (6.2; 6.6)	<0.001
*p*-value	<0.001	<0.001	<0.001	<0.001	<0.001	
Acute diabetes complications						
Yes	6.1 (5.8; 6.4)	6.8 (5.8; 7.7)	5.9 (5.4; 6.4)	6.4 (5.8; 7.0)	5.7 (5.3; 6.2)	0.083
No	7.4 (7.3; 7.5)	7.4 (7.2; 7.6)	7.6 (7.4; 7.8)	7.4 (7.2; 7.6)	7.1 (7.0; 7.3)	0.003
*p*-value	<0.001	0.377	0.001	0.128	0.002	

* Presented as mean (95% Confidence Interval). CVD, cardiovascular diseases.

**Table 3 ijerph-19-10035-t003:** Adjusted median cost (95% CI) per episode of hospitalization in the analyzed period (EUR).

Median Cost per Episode, EUR	All Sample N = 28,055	2015 N = 6899	2016 N = 6756	2017 N = 7025	2018 N = 7408	*p*-Value for Trend
Overall	596.5 (347.5; 994.1)	591.7 (338.1; 977.2)	621.9 (375.7; 1031.0)	596.0 (348.3; 990.4)	577.4 (336.1; 971.3)	<0.001
Type of diabetes						
Type 1	410.3 (258.9; 675.1)	395.8 (254.5; 649.5)	436.4 (273.6; 681.8)	478.0 (285.6; 822.2)	389.4 (231.5; 606.5)	<0.001
Type 2	604.5 (355.2; 1009.1)	602.7 (344.3; 992.1)	631.6 (388.0; 1046.0)	600.4 (353.2; 1000.8)	589.2 (343.4; 994.2)	<0.001
Specific forms	659.5 (387.9; 1111.6)	691.8 (380.4; 1268.3)	716.0 (410.4; 1219.2)	687.4 (389.2; 1261.2)	590.3 (373.3; 875.3)	0.142
*p*-value	<0.001	<0.001	<0.001	<0.001	<0.001	
By sex						
Women	606.9 (365.3; 992.7)	596.3 (344.2; 972.1)	638.0 (407.5; 1049.6)	612.5 (370.4; 989.2)	580.0 (345.4; 948.6)	<0.001
Men	581.8 (333.5; 996.5)	585.0 (332.0; 985.3)	595.9 (345.0; 1017.4)	577.2 (329.5; 992.4)	570.8 (324.3; 992.8)	0.047
*p*-value	<0.001	0.255	<0.001	0.001	0.417	
By age groups						
18–40 years	402.8 (252.6; 632.1)	344.4 (242.7; 552.9)	426.2 (272.4; 639.2)	415.2 (270.1; 799.4)	393.3 (232.3; 600.2)	0.003
40–65 years	542.9 (320.8; 889.8)	542.5 (318.0; 876.8)	568.6 (342.4; 929.3)	545.8 (322.1; 894.6)	514.0 (303.6; 861.2)	<0.001
≥65 years	665.3 (408.7; 1100.4)	670.7 (405.4; 1082.8)	705.5 (428.6; 1157.0)	655.0 (404.8; 1094.4)	639.9 (395.3; 1074.9)	<0.001
*p*-value	<0.001	<0.001	<0.001	<0.001	<0.001	
Diabetic retinopathy						
Yes	365.7 (149.2; 608.0)	368.0 (171.4; 620.1)	414.3 (167.6; 622.6)	368.0 (146.2; 618.4)	297.2 (96.4; 551.4)	<0.001
No	638.8 (389.7; 1072.5)	642.4 (374.6; 1062.0)	667.0 (416.8; 1124.4)	632.8 (386.4; 1068.2)	614.4 (378.4; 1048.5)	<0.001
*p*-value	<0.001	<0.001	<0.001	<0.001	<0.001	
Diabetic neuropathy						
Yes	545.1 (364.1; 787.3)	542.2 (346.7; 775.5)	560.6 (393.8; 795.4)	550.0 (370.8; 806.0)	525.0 (357.4; 768.5)	0.019
No	621.0 (343.1; 1063.8)	622.0 (334.1; 1054.2)	653.1 (371.7; 1112.9)	619.5 (342.9; 1062.7)	597.0 (330.9; 1035.6)	<0.001
*p*-value	<0.001	<0.001	<0.001	<0.001	<0.001	
Chronic kidney disease						
Yes	649.5 (429.5; 1046.4)	631.2 (407.0; 981.2)	672.3 (466.6; 1086.4)	651.4 (441.7; 1045.9)	641.2 (414.7; 1048.7)	0.004
No	582.7 (336.0; 980.8)	583.9 (328.8; 974.8)	606.1 (360.2; 1020.5)	581.3 (337.7; 973.0)	559.6 (323.8; 953.7)	<0.001
*p*-value	<0.001	<0.001	<0.001	<0.001	<0.001	
Foot ulcerations						
Yes	810.9 (587.7; 1325.5)	815.4 (592.0; 1464.0)	814.6 (605.5; 1293.8)	797.2 (559.8; 1274.3)	822.2 (574.3; 1290.5)	0.351
No	587.1 (341.7; 980.7)	580.5 (332.3; 950.0)	610.3 (368.7; 1021.2)	589.2 (342.8; 981.9)	568.1 (329.4; 957.7)	<0.001
*p*-value	<0.001	<0.001	<0.001	<0.001	<0.001	
CVD						
Yes	683.4 (445.4; 1094.0)	673.8 (435.7; 1076.7)	712.3 (470.0; 1110.7)	686.9 (449.0; 1094.3)	666.6 (440.3; 1091.9)	0.017
No	529.0 (295.9; 899.1)	515.0 (276.7; 889.0)	567.3 (330.1; 954.2)	532.7 (298.3; 897.1)	500.1 (274.2; 853.3)	<0.001
*p*-value	<0.001	<0.001	<0.001	<0.001	<0.001	
Acute diabetes complications						
Yes	479.4 (304.4; 720.1)	495.0 (310.9; 792.0)	482.3 (318.1; 693.2)	519.4 (319.7; 876.5)	448.7 (299.1; 616.1)	0.050
No	600.2 (349.5; 1003.8)	593.5 (338.6; 984.1)	626.4 (378.4; 1046.3)	597.8 (349.2; 995.1)	582.5 (338.5; 987.0)	<0.001
*p*-value	<0.001	0.007	<0.001	0.072	<0.001	

CI, confidence interval; CVD, cardiovascular disease (peripheral arteriopathy with ulceration not included); N, number of episodes of hospitalizations.

**Table 4 ijerph-19-10035-t004:** Predictors of CPI-adjusted costs and length of stay per episode of hospitalization.

Predictor	Adjusted Average Costs per Episode of Hospitalization		Length of Stay per Episode of Hospitalization	
	Standardized β Coefficient	*p*-Value	Standardized β Coefficient	*p*-Value
Age group	0.027	<0.001	0.054	<0.001
Sex (men vs. women)	0.015	0.015	0.012	0.051
Diabetes type (specific forms vs. type 2 vs. type 1)	0.000	0.949	0.008	0.213
Acute diabetes complications (yes vs. no)	−0.019	0.002	−0.012	0.040
Diabetic retinopathy (yes vs. no)	−0.085	<0.001	−0.120	<0.001
Diabetic neuropathy (yes vs. no)	−0.032	<0.001	−0.003	0.610
Chronic kidney disease (yes vs. no)	0.016	0.010	0.037	<0.001
Foot ulcerations (yes vs. no)	0.030	<0.001	0.099	<0.001
CVD (yes vs. no)	0.017	0.008	0.027	<0.001

CVD, cardiovascular disease.

## Data Availability

The data presented in this study are available on request from the corresponding author.

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
