# Peer review of "Length of Hospital Stay, Hospitalization Costs, and Their Drivers in Adults with Diabetes in the Romanian Public Hospital System"

_ijerph, 2022, doi:10.3390/ijerph191610035_

Round 1
Reviewer 1 Report
There are several limitations to this paper.
1. English language needs improvement.
2. Methods are week. No justification provided for not including secondary care hospitals. No cost smoothening analysis is done. At several places we need median rather than average cost with range.
3. Objectives are not adequately defined and no hypothesis were generated.
4. The database included on 18+ population as a result many children and teenagers suffering from Type 1 will be eliminated. This is a big limitations. In fact no analysis is needed to compare Type 1 with Type 2.
5. Wherever costs of hospitalisation compared across various countries better to use International $ (PPP) which reflect on purchasing power. Therefore in the discussion and Introduction sections such statistics comparison is useless.
6. Better to provide all detailed ICDS disease codes considered for analysis in the appendix.
7. Some of the inferences drawn in the discussions are not correct.
8. Finally, for international readers it is better to provide context of Romanian healthcare system, financing and insurance coverage specifically for diabetes care.

Author Response
There are several limitations to this paper.
- English language needs improvement.
Response:
We thank the reviewer for taking the time to review our manuscript. We have reviewed the entire manuscript and corrected the English language as requested.
We have also taken into account all comments provided in the report and in the pdf manuscript uploaded by the Reviewer.
- Methods are week. No justification provided for not including secondary care hospitals. No cost smoothening analysis is done. At several places we need median rather than average cost with range.
Response:
A justification for including the tertiary care hospital have been included in the Methods section page 3. Also, the average costs/episode of hospitalization have been replaced with median costs and quartiles 1 and 3 (Results section, pages 6 and 7). The description of ICD codes has been moved in the Appendix. We acknowledge that data smoothing would improve a forecast analysis. However, here we aimed to provide a report of previous costs and not to model future costs of hospitalization of patients with diabetes in Romanian public hospitals. Thus, we avoided smoothing data for statistical analysis.
- Objectives are not adequately defined and no hypothesis were generated.
Response:
We have defined the hypothesis of the research and the objectives have been clarified, as follows (Introduction section, page 2):
In this context, we hypothesized that inhospital care of persons with diabetes accounts for a high proportion of the hospital expenses. In the research presented here, we aimed to calculate the costs associated with the hospitalization of adults with diabetes, and to identify the main drivers of these costs.
- The database included on 18+ population as a result many children and teenagers suffering from Type 1 will be eliminated. This is a big limitations. In fact no analysis is needed to compare Type 1 with Type 2.
Response:
We thank the reviewer for this comment. The Emergency Clinical County Hospital is admitting mainly adults and there is no department for children. Thus, children and adolescents are rarely admitted in this hospital. We have updated the title and objectives of the manuscript to clarify that this cost analysis was performed only in adults.
- Wherever costs of hospitalization compared across various countries better to use International $ (PPP) which reflect on purchasing power. Therefore in the discussion and Introduction sections such statistics comparison is useless.
Response:
In the published literature presenting the hospitalization costs of diabetes and other diseases are used USD, BGP or EUR. Indeed, this makes the comparison among studies in various countries difficult. We have chosen to convert the costs from Romanian Leu to EUR so we can compare the costs with those reported in other European countries. The usage of EUR may also allow the policy makers in Romania to use our results to forecast costs and inform future decision in financing diabetes care in Romania. We have adjusted the costs with the annual inflation rates calculated by the Romanian National Statistics Institute using the Consumer Price Index to account for changes in the purchasing power and taking into account the average exchange rate of the Romanian National Bank, for each year of the period 2015–2018. We chose as base year the last year. Such an approach is widely used in similar studies which have been added as reference in the Methods section, page 4.
We have also improved the Discussion section by updating the references on the diabetes costs per episode of hospitalization. Now comparisons are performed with figures expressed in EUR as in our manuscript.
- Better to provide all detailed ICDS disease codes considered for analysis in the appendix.
Response:
As requested, we have moved the description of ICD codes in the Appendix.
- 7. Some of the inferences drawn in the discussions are not correct.
Response:
We have improved the paragraphs highlighted in the Discussion section (page 14) of the pdf attached to document with Reviewer’s comments. We have removed from the Discussion section the comparison of our results on the median cost/episode of hospitalization with previous figures expressed in USD or GBP. These were replaced with comparisons with figures on costs expressed in EUR. We have also changed the paragraph on the effect of length of hospitalization on costs. Now it states that length of hospitalization has an incremental effect on costs, as Reviewer kindly suggested.
- Finally, for international readers it is better to provide context of Romanian healthcare system, financing and insurance coverage specifically for diabetes care.
Response:
We have added in the Methods section, page 4, a paragraph describing the Romania healthcare system and how the therapy of patients with diabetes is covered.
Reviewer 2 Report
Diabetes mellitus is clearly a disease that needs attention in terms of cost and the increasing number of patients worldwide.
Therefore, this study is interesting and significant because it examines the factors that influence the length of hospital stay and the cost of healthcare for diabetes.
Overall, there is nothing to point out, but minor remarks and comments are made as follows:
・L171 "96% Confidence Interval". Normally, 95% confidence intervals are used, but what is the intention behind the 96%?
The following is a comment and is not a request for a response. I hope you will find it informative.
 This time, costs (L122-143) are considered together. I think that incorporating each cost factor as an explanatory variable in a multivariate analysis (regression analysis) will make the picture of hospital admissions clearer. It may be possible to estimate whether the cost increase is due to drugs, devices or the length of hospital stay for follow-up.
Please consider this.
Author Response
Reviewer 2
Diabetes mellitus is clearly a disease that needs attention in terms of cost and the increasing number of patients worldwide.
Therefore, this study is interesting and significant because it examines the factors that influence the length of hospital stay and the cost of healthcare for diabetes.
Overall, there is nothing to point out, but minor remarks and comments are made as follows:
・L171 "96% Confidence Interval". Normally, 95% confidence intervals are used, but what is the intention behind the 96%?
Response:
We have corrected the typo. We have used 95% confidence intervals.
The following is a comment and is not a request for a response. I hope you will find it informative.
 This time, costs (L122-143) are considered together. I think that incorporating each cost factor as an explanatory variable in a multivariate analysis (regression analysis) will make the picture of hospital admissions clearer. It may be possible to estimate whether the cost increase is due to drugs, devices or the length of hospital stay for follow-up.
Please consider this.
Response:
We thank the reviewer for this comment. Indeed, including each cost factor in a multivariate regression analysis would improve the understanding of drivers of costs. However, we were provided the total amount that included all costs and not with separate costs for bed, food, drugs, medical supplies, and services. Thus, we cannot perform the multivariate analysis with these costs included separately. This is listed as a study limitation in the Discussion section, page 15.
Reviewer 3 Report
In this study, the authors report results from a Romanian single-center retrospective analysis of hospitalization costs and associated factors in patients with diabetes. The results are in line with the available literature and do not offer any major novelties, however there is only little data on the studied topic available in Romania, and the report might thus be of interest for public health institutions. The paper is well-written. The study contains a large sample size and adequate statistical analysis, presentation and discussion.
There are three points that should be addressed in the discussion before publication:
1.) The references for costs in other European countries are outdated and thus not ideally suited for comparison with this study (ref. 14-17). Perhaps the authors can find more up-to-date numbers for comparison in the discussion section.
2.) An important, but omitted limitation lies in the single-center study design - the authors should discuss this limitation and its implications.
3.) Another important limitation lies in the lack of sociodemographic data, specifically smoking status with regard to CVD - some significant effects on the outcomes are to be expected with stratification by smoking status. The authors should expand on this issue and discuss it in a dedicated separate paragraph.
Additionally, there are some minor grammatical/stylistic recommendations:
Materials L90: We retrospectively analyzed …
Materials L142: X-Ray
Results L187: Do not split table 1 between two pages
Results L203: A similar evolution was seen in the length…
Results L220: (…), while the highest number (…) was registered
Discussion L28: The American Diabetes Association …
Discussion L99: (…) ‘were available for this analysis’ at the end of the sentence.
Author Response
Reviewer 3
In this study, the authors report results from a Romanian single-center retrospective analysis of hospitalization costs and associated factors in patients with diabetes. The results are in line with the available literature and do not offer any major novelties, however there is only little data on the studied topic available in Romania, and the report might thus be of interest for public health institutions. The paper is well-written. The study contains a large sample size and adequate statistical analysis, presentation and discussion.
There are three points that should be addressed in the discussion before publication:
- The references for costs in other European countries are outdated and thus not ideally suited for comparison with this study (ref. 14-17). Perhaps the authors can find more up-to-date numbers for comparison in the discussion section.
Response:
We have replaced the outdated references for costs in other European countries with numbers provided by International Diabetes Federation in the 10th Diabetes Atlas published in 2021.
2.) An important, but omitted limitation lies in the single-center study design - the authors should discuss this limitation and its implications.
Response:
We have added the single center design as a study limitation in the Discussion section, page 15.
3.) Another important limitation lies in the lack of sociodemographic data, specifically smoking status with regard to CVD - some significant effects on the outcomes are to be expected with stratification by smoking status. The authors should expand on this issue and discuss it in a dedicated separate paragraph.
Response:
We have added in the Discussion section, page 15, the lack of sociodemographic data as a study limitation. We have also added a brief discussion on the influence of smoking status on the hospitalization costs.
Additionally, there are some minor grammatical/stylistic recommendations:
Materials L90: We retrospectively analyzed …
Materials L142: X-Ray
Results L187: Do not split table 1 between two pages
Results L203: A similar evolution was seen in the length…
Results L220: (…), while the highest number (…) was registered
Discussion L28: The American Diabetes Association …
Discussion L99: (…) ‘were available for this analysis’ at the end of the sentence.
Response:
We thank the reviewer for these recommendations. We have corrected all grammatical/stylistic errors.
Round 2
Reviewer 1 Report
Thank you for accepting my suggestions and revised the document. I think a limitation paragraph may be added after the discussion. The hypotheses have not been identified and described. In other words what type of relationships you would expect from the data analysis when compared with the existing evidence.
Author Response
Reviewer 1
Thank you for accepting my suggestions and revised the document. I think a limitation paragraph may be added after the discussion.
Response
A study limitations paragraph is already present at the end of the Discussion section, page 12, lines 96-108.
The hypotheses have not been identified and described. In other words what type of relationships you would expect from the data analysis when compared with the existing evidence.
Response:
As suggested, we have updated the hypothesis. Now it reads (page 2, lines 86-88):
In this context, we hypothesized that in Romania as in other countries, the costs of hospitalization of patients with diabetes are high and hospital care of these patients accounts for a high proportion of the hospital expenses.
Also, based on the feedback provided in the journal’s system, a review was performed by a proficient English language user and we corrected the English language throughout the manuscript. All changes are provided in track changes.